# Hearing and Language Skills in Children Using Hearing Aids: Experimental Intervention Study

**DOI:** 10.3390/jpm14040372

**Published:** 2024-03-30

**Authors:** Luana Speck Polli Burigo, Anna Quialheiro, Karina Mary de Paiva, Thaiana Vargas dos Santos, Luciele Kauana Woide, Luciana Berwanger Cigana, Janaina Massignani, Patricia Haas

**Affiliations:** 1PPGFONO-UFSC, Eng. Agrônomo Andrei Cristian Ferreira, Bairro, Trindade, CEP, Florianópolis 88040-900, SC, Brazil; luaspeck@gmail.com; 2CESPU, José Antônio Vidal Street, 81, Defesa, 4760049 Famalicão, Portugal; annaqas@gmail.com; 3Department of Speech Therapy, Universidade Federal de Santa Catarina-UFSC, Eng. Agrônomo Andrei Cristian Ferreira, Bairro, Trindade, CEP, Florianópolis 88040-900, SC, Brazil; kmvianna@gmail.com (K.M.d.P.); thaiana203@gmail.com (T.V.d.S.); lucielekauana@gmail.com (L.K.W.); 4Instituto Otovida, Florianopolis, Brazil. Av. Gov. Ivo Silveira, 3861—Capoeiras, Florianópolis 88085-002, SC, Brazil; lbcigana@gmail.com (L.B.C.); janainamassignani@gmail.com (J.M.); 5Speech Therapy Department, Universidade Federal da Fronteira Sul—UFFS, Campus Chapecó, CEP, Chapecó 89815-899, SC, Brazil

**Keywords:** hearing loss, intervention, listening skills, personal sound amplification device

## Abstract

Introduction: Hearing loss in childhood compromises a child’s auditory, linguistic, and social skill development. Stimulation and early intervention through therapy and the use of personal sound amplification devices (PSAPs) are important for improving communication. Purpose: To verify the effectiveness of speech therapy intervention on the auditory and linguistic skills of Brazilian children aged between 6 and 8 years using PSAPs. Methods: Experimental study analyzing the intervention process in children aged between 6 and 8 years with mild to severe bilateral hearing loss and prelingual deafness who are PSAP users. Diagnostic information was analyzed, and assessments and interventions were carried out using the Glendonald Auditory Screening Procedure (GASP), a phoneme discrimination test with figures (TFDF), an expressive language category classification test, and an Infant-Toddler Meaningful Auditory Integration Scale (IT-MAIS) questionnaire. Results: Sixteen children participated in the study; they were divided into a control group (CG) of six children and an intervention group (IG) of ten children. All research subjects underwent two protocol application sessions, and the IG underwent six speech therapy intervention sessions. In the IT-MAIS, the CG had a 9% increase in score, and the IG had an increase of 3% after intervention. The TFDF obtained a 5% increase in the IG in terms of phonemic discrimination ability. The expressive language category classification tests and GASP were considered not sensitive enough to modify the parameters of auditory and linguistic skills. Conclusions: The study found a significant improvement amongst the IG in the TFDF protocol and an increase in IT-MAIS scores in both groups.

## 1. Introduction

Hearing skills play a crucial role in the development of hearing-impaired children who use personal sound amplification devices (PSAPs). According to the International Pediatric Audiology Association (IPAD), early and personalized interventions are essential in promoting these children’s hearing development. The International Audiology Association (IPAA) emphasizes the importance of continuous auditory stimulation, proper fitting of hearing aids, and regular professional monitoring and the use of multimodal approaches that combine auditory and visual information. Furthermore, the IPAA highlights the need for an enriched language environment, with meaningful social interactions and access to support services, to ensure the full development of these children’s listening and language skills [1].

Research [2] analyzed the speech of 15 children and adolescents with moderate bilateral sensorineural hearing loss who have used PSAPs for more than 4 years and underwent auditory rehabilitation. They observed that the individuals who participated in their research achieved good speech intelligibility, proving that while oral communication is a challenge for children with hearing impairment, if their auditory skills are stimulated, their speech production becomes more adequate and effective than with the use of a PSAP alone.

Oliveira et al. [3] carried out a language development study in children with hearing loss who used a PSAP and evaluated their oral and written skills, as well as changes in these skills. Hearing loss causes harm to language development; the greater the degree of hearing impairment, the greater the difficulty in perceiving and discriminating speech, and the greater the deficits in language. The scientific productions analyzed in this study revealed a wide variety of tests have been used to evaluate language. However, they observed that there are no specific standard protocols to analyze the language development of children with hearing impairments.

The objective of this study was to verify the effectiveness of speech therapy intervention, focusing on the auditory and linguistic skills of Brazilian children aged between 6 and 8 years who use a digital sound amplification device for at least 4 h per day.

## 2. Methodology

Type of study: Experimental study with a control group with children aged between 6 and 8 years who attended the Hearing Health Reference Service (SASA) of the Brazilian Unified Health System (SUS) in the State of Santa Catarina between January and April 2023.

Inclusion criteria: For the initial study screening, diagnostic information was collected from audiological exams and auditory rehabilitation of all children aged between 6 and 8 years with mild to severe bilateral hearing loss and prelingual deafness, who use a personal sound amplification device (PSAP), and who were admitted during the study period and referred for evaluation and intervention.

Exclusion criteria: Children with neurological changes, autism spectrum disorder (ASD), or other neurodevelopmental disorders or syndromes, or who had already undergone an intervention to adapt their hearing aid prior to the present study, were excluded from this research.

Population: After selecting the children, their guardians were contacted via telephone and an invitation to participate in the study was offered. After recruitment, consent to participate in the study was given by the guardian signing a consent form. Following this, anamnesis and exams, including questions about hearing difficulty, physical and laboratory examinations, associated diseases, clinical history, and whether the child is a PSAP user, were obtained from the SASA database. After obtaining the data, the research agenda was structured according to compatibility with the family and the availability of municipality transport for weekly care logistics.

Protocols: Protocol applications were carried out at the beginning and end of the intervention for the intervention group (IG) and control group (CG). The protocols consist of the assessment of auditory and linguistic skills and are approved instruments for this age group and application in the Brazilian population. Four protocols were used: IT-MAIS, TFDF, GASP, and a language category classification test.

The Infant-Toddler Meaningful Auditory Integration Scale (IT-MAIS) questionnaire evaluates responses to speech sounds and environmental sounds exclusively through the auditory sensory pathway. It consists of open and closed questions whose main goal is to evaluate speech auditory perception. The assessment was based on information provided by the child’s parents or guardians in response to 10 questions that assessed three main areas: vocalization behavior, attention to sounds, and meaning attribution based on the sound.

The phoneme discrimination test with a figure (TFDF) was applied to evaluate the phonemic discrimination ability through minimal pairs, and the final score was classified as adequate or altered, according to age, as per criteria proposed in the literature. It is a test that incorporates 30 minimum pairs (60 words), represented by pictures and organized into presentation cards. To explain the test tasks, four demonstration items are part of the test. The 30 minimum pairs were organized into 40 presentations, of which 30 contain two different words and 10 contain two identical words. The TFDF tasks consist of listening to two words (which can be different or the same) and pointing to the card containing the pictures that represent them.

The Glendonald Auditory Screening Procedure (GASP) is a speech perception test composed of three levels and five tests that assess the ability to detect language sounds, vowel discrimination, length discrimination, word recognition, and sentence comprehension with the use of an individual sound amplification device. To verify the development of auditory recognition in a closed set and listening comprehension, test 5 of the GASP was analyzed, which assesses the ability of auditory recognition in a closed set and is composed of monosyllabic, disyllabic, trisyllabic, and polysyllable words, totaling 12 words presented through figures.

To classify expressive language categories, the Brazilian version of the American Reynell Developmental Language Scales (RDLS) instrument was applied. This instrument is composed of two scales for evaluating oral language, the verbal comprehension scale and the expression scale. These scales aim to compare the relationship between oral comprehension performance and oral expression performance.

**Intervention:** The intervention consisted of six speech therapy sessions lasting 60 min each (six sessions is the minimum to stimulate the skills proposed in the study). The CG was instructed to stimulate auditory skills of sound detection and discrimination in the family environment, as well as receiving parental guidance to promote bonding with children.

The speech therapy sessions aimed to stimulate the auditory skills of children with prelingual deafness. Each intervention had a specific and evolutionary purpose, from developing reciprocity of action and communication to developing the functionality of auditory and language skills for daily-living activities (Figure 1). After the end of the intervention, children in the IG and CG groups were reassessed using the same initial protocols.

**Data analysis:** Data were organized in Microsoft Excel^®^ and later exported and analyzed using StataMP^®^ software, version 14.0 (StataCorp., College Station, TX, USA). To characterize the sample, descriptive analyzes were presented, with absolute and relative frequencies and their respective 95% confidence intervals (95% CI). As it was a small sample, for pre- and post-intervention analysis, Student’s *t*-test was used for non-parametric samples. Comparative analysis was carried out between the intervention and control groups using Student’s *t*-test for independent and non-parametric samples. A comparative analysis was also carried out using Student’s *t*-test, for paired and independent samples, in order to obtain the mean and 95% CI. For statistical significance, a *p*-value of less than 0.05 was considered. Values obtained via Student’s *t*-test were used when significance was verified in the same way as the non-parametric test.

This study was approved by the Research Ethics Committee (CEP) of the Universidade Federal de Santa Catarina. CAAE: 64315322.7.0000.0121. As the sample consisted of subjects under 18 years of age, the consent form was signed by the legal guardian.

## 3. Results

Sixteen children participated in this study (IG (n = 10) and CG (n = 6)). The mean age of participants was 6 years, with birth period of between 2005 and 2006. The CG had four withdrawals due to difficulties with the municipality’s transport logistics. Two sessions were required to apply protocols before the intervention period. Children from the IG underwent six speech therapy sessions between February and April 2023.

Most children lived in the municipality of São José, followed by Florianópolis, and Jaraguá do Sul. Regarding education, most of them were attending their 1st year in Brazilian elementary school. Regarding PSAP adaptation, the majority underwent adaptation in 2022, followed by 25% in 2020 and 2021 (Table 1).

When applying the IT-MAIS questionnaire, we observed that the scores of the CG increased with reduced intensity. The IG did not present significant changes in scores from the first application, before the intervention, to the protocol application at the end of the six interventions. We observed that the CG had a very low score before the intervention period, presenting significant differences between the CG and IG, thus making it impossible to compare the two groups after the intervention. The IG started the intervention with a high score, and even with the intervention sessions on auditory and linguistic skills, while an increase in the score was observed, it was not significant (Table 2).

The CG median was higher than the IG median in the pre-intervention period assessment when considering the difference in score for the IT-MAIS questionnaire (Figure 2).

Regarding the TFDF, the results in the CG did not show a significant difference after the intervention period. The IG showed a significant improvement between the first and second application, due to an increase in the mean score (from 23 to 28.1 points, on average) in the TFDF (Table 3).

When analyzing the difference between the pre- and post-intervention scores of both groups, it is clear that even though the CG median is higher than the IG median, the IG presented higher values than the CG (when observing the interquartile range from Figure 3).

The GASP expressive language and speech perception tests were non-sensitive instruments to differentiate the children’s development in this study. In the expressive language classification test, only one child changed from category 4 to 5, the others remained in the same category. Likewise, using the GASP instrument, when observing the initial values, children in both groups maintained their respective classification.

To analyze the follow-up time (the difference in dates between the first and second sessions of the protocol application), the CG had a mean interval of 38 days between the first and second evaluation, while the IG had a mean interval of 55 days.

## 4. Discussion

In line with the purpose of this study, the results of applying the protocols and speech therapy intervention, used as a proposal for developing the auditory and linguistic skills of this population, indicate that the stimulation of skills linked to the scientific protocols used constitutes a positive process in the development of the assessed skills.

Regarding the IT-MAIS protocol, when analyzed by group (IG and CG), the CG showed a significant increase in the mean score, while the IG also showed an increase in the mean score, albeit not significant, post-intervention. All questions used in the IT-MAIS questionnaire are instruments for reflection regarding the child’s use of a PSAP, as well as regarding its handling and levels of use in the family environment. All those responsible, when answering the IT-MAIS questionnaire twice, were guided and led to new approaches and management strategies regarding the PSAP and dialogue with the child. The study did not show a significant increase in the IG score after intervention, but both groups responded very similarly.

The study [4] consisted of a sample of twelve participants divided into two groups: G1 with children from 6 months to 2 years of age and G2 with children from 2 to 4 years of age. The two groups were analyzed by reading and interpreting the IT-MAIS at the beginning and end of the intervention. The IT-MAIS is a questionnaire used with children who are in their first year of PSAP adaptation to validate the procedures carried out in rehabilitation. The first results obtained were about the mean daily use of PSAP, which was below what was expected for the participants’ age. In two patients, there was a decrease in the post-intervention questionnaire score, confirming that the mean number of hours of PSAP use had decreased. The best score responses in this study were observed in users with mild or moderate hearing loss.

The two groups (CG and IG) did not present relevant results for the GASP protocol or the expressive language category test. The GASP is a test that evaluates the ability of auditory recognition in a closed set and is composed of monosyllabic, disyllabic, trisyllabic, and polysyllabic words. It totals twelve words presented through pictures, and the expressive language category test, which is composed of two scales for evaluating oral language: the verbal comprehension scale and the expression scale. The two tests were not sensitive to the changes that occurred between the groups and after intervention in the IG, and the analysis showed that only one participant in the IG showed a change from category 4 to category 5; the others remained in the same category.

In a study [5] the GASP speech perception assessment test was used, which assesses detection skills, vowel discrimination, extension perception, auditory recognition and sentence comprehension. The TFDF test, which contains figures to assess phonemic discrimination in minimal pairs and temporal processing, was also used. One hundred and ten children aged between 6 and 10 years with mild to profound hearing loss, and who were PSAP users, took part. Degree of hearing loss and PSAP adaptation time were evaluated. Regardless of the degree of hearing loss, the diagnosis and necessary interventions occurred late, compromising the linguistic and auditory skills of these children.

As for the TFDF protocol, which assesses phoneme discrimination ability through minimal pairs, a significant difference was found, with an improvement in the score of the IG, which went from 23 to a score of 28.1 on average, while in the CG there was no significant difference.

The threshold-equalizing dose frequency (TFDF) test is a hearing assessment tool frequently applied to hearing-impaired children and hearing aid users. This test aims to determine the hearing threshold and equalize auditory responses according to the sound frequency. It allows an accurate assessment of children’s hearing abilities, helping to properly adjust hearing aids and identify possible hearing difficulties in different frequency ranges. The TFDF is a valuable resource for healthcare professionals to monitor and develop personalized auditory rehabilitation strategies for children with hearing impairment [6]

The IG intervention sessions were based on goals that focused on developing reciprocity of action and communication and promoting attention skills and perception of self and others, as well as intentional communication. All of these skills were stimulated through playful activities, scientifically supported and with therapeutic resources, so that, in the long term, IG children achieved better percentages in auditory and language skills.

The hearing aid is an individual electronic device, and each patient has a different audiological diagnosis and needs, with specific sound amplification gain requirements. However, even with all the technology invested in PSAPs, there are flaws in the software and systems built into the PSAP that interfere with the gain, such as interference from loud noises and batteries that discharge quickly [7]

The results reiterate the importance of children with hearing loss and PSAP users attending an intervention to stimulate auditory skills. Such actions, in the intervention context, have repercussions on daily family life, with an important positive impact on children’s communication, providing support for therapeutic guidance for a more effective intervention from the perspective of comprehensive and personalized care.

Therapy for PSAP users plays a fundamental role in the auditory rehabilitation process and in improving auditory skills. According to the American Speech–Language–Hearing Association (ASHA), an international reference in the hearing field, hearing aid therapy can include a variety of approaches, such as auditory training, verbal auditory therapy, and communication strategies. These therapies aim to improve auditory perception and processing, promoting better speech understanding, sound localization, and auditory discrimination. ASHA also emphasizes the importance of therapy in conjunction with proper fitting and calibration of hearing aids, as well as ongoing support from specialized professionals. By following these international guidelines, therapy for hearing aid users can help maximize hearing benefits and improve their quality of life [8].

Hearing rehabilitation for PSAP users is a practice that is little sought-after, often for economic, but also for cultural and social, reasons. One of the aims of this rehabilitation is to stimulate the auditory abilities of detection, discrimination, localization, auditory recognition, and auditory comprehension. The importance of developing neuroplasticity is highlighted, so that children who use electronic devices learn the meaning of the sounds they hear and obtain good results in the performance of auditory skills, spoken language, speech intelligibility, and quality of life [9].

Ref. [10] research (2022) confirmed that aural intervention achieves progress in spoken language and speech intelligibility in all groups that were analyzed in the study. This method is considered an intervention that helps children discover, understand, and increase their linguistic repertoire, and is an option for children with hearing impairment and PSAP or CI users. Through the effective use of the electronic device, strategies and practices in intervention, family support, and school participation, it is believed that the effectiveness of communication in a child’s development emerges more naturally in listening and speaking.

Studies that investigated the progress in the acquisition of spoken language in children with hearing loss who use PSAPs, and who are included in therapy based on gradually developing auditory skills, demonstrate satisfactory advances when compared to their hearing peers [11,12].

Advances in the acquisition of spoken language by children with hearing loss and hearing aid users have been a focus of international research and practice. Studies and guidelines, such as those published by the Joint Committee on Infant Hearing (JCIH), have contributed to this process by promoting effective strategies. Through the use of hearing technologies, such as hearing aids and cochlear implants, combined with specialized interventions, such as verbal auditory therapy, these children have demonstrated significant progress in the development of spoken language. The JCIH highlights the importance of early diagnosis, early intervention, and ongoing support, aiming to maximize the communication and language opportunities of these children from the first years of life. With adequate monitoring and access to specialized services, it is possible to provide satisfactory linguistic development, allowing these children to reach their full communicative potential [1].

For language development to occur, it is important that each individual is analyzed and guided individually, according to their hearing needs with a PSAP. Positive results arise when treatment and intervention are designed directly for that child and their social environment. Children with hearing loss do not fully capture verbal and non-verbal sounds. By capturing sounds via a PSAP, access to speech and environmental sounds is provided and, when well stimulated, children obtain better auditory and linguistic skills. The development of these skills occurs if the PSAP user uses the device for at least 10 h/day; thus, the child retains direct and indirect information that influences the increase in their auditory and linguistic repertoire [13].

A limitation of this study is the lack of protocols that are based on scientific evidence, which prove in advance the reliability of taking risks and can demonstrate significant changes in the auditory and linguistic skills of deaf children.

This study was based on the literature and, with protocols used for children with delayed language development and the lack of a specific validated protocol, it was necessary to use four questionnaires to fulfill the study’s purpose. However, the study demonstrated that the TFDF and IT-MAIS protocols showed the most significant change during the proposed intervention. Thus, this structure, with the necessary therapeutic objectives, has led children with hearing loss to better ways of communicating with, and learning about, the world they live in.

## 5. Conclusions

This study found that an intervention with six stimulation therapy sessions in the studied population tended to improve the communication skills of children who use a PSAP. The IT-MAIS and TFDF auditory and linguistic skills assessment protocols presented important applicability in the verification and conduct of research follow-up. The GASP and expressive language category test protocols did not detect improvements in the auditory and linguistic skills of children targeted in this study.

Considering this context, in order to guarantee the sustainability of public policies through the provision of specialized care for children who use a PSAP, the authors hope that the results of this study might influence the course of the SUS’s investment policy. Strategies such as welcoming the patient and offering specialized care to stimulate hearing skills, in addition to more effective healthcare, highlight several concerns of parents regarding the implications of communication changes for their children’s relationships with other people and their consequences for their quality of life and future.

## Figures and Tables

**Figure 1 jpm-14-00372-f001:**
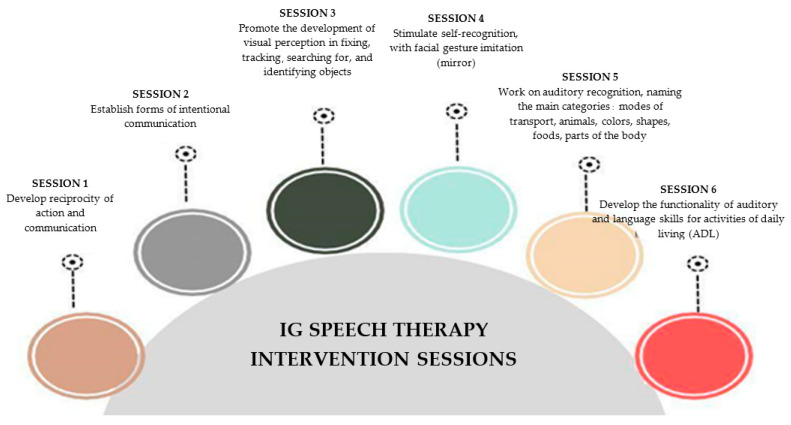
Description of sessions for the intervention group (IG).

**Figure 2 jpm-14-00372-f002:**
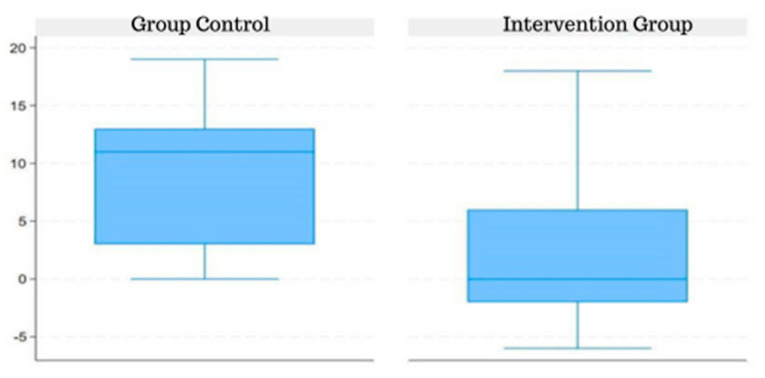
Difference in the ITMAIS test score pre- and post-intervention in the CG and IG.

**Figure 3 jpm-14-00372-f003:**
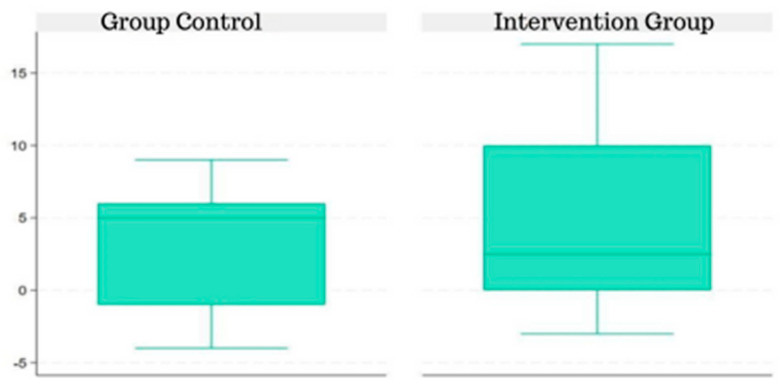
Difference in the TFDF test score pre- and post-intervention in the CG and IG.

**Table 1 jpm-14-00372-t001:** Sample description according to year of birth, age, city, education, and PSAP adaptation. Florianópolis, SC, 2023 (n = 16).

Variable	IG (n)	CG (n)	Total (n)	%
Age				
6 years	8	2	10	62%
7 years	2	2	4	25%
8 years	1	1	2	12%
Sex				
Male	7	6	13	81%
Female	3	0	3	18%
City				
Florianópolis	3	0	3	18%
São José	4	0	4	25%
Palhoça	2	0	2	12%
Jaraguá do Sul	1	2	3	18%
São Bento do Sul	0	1	1	6%
Capão Alto	0	1	1	6%
Brusque	0	1	1	6%
São Joaquim	0	1	1	6%
Education (Elementary School)				
1st year	8	2	10	62%
2nd year	1	3	4	25%
3rd year	1	1	2	12%
PSAP adaptation (year)				
2020	1	3	4	25%
2021	3	1	4	25%
2022	6	2	8	50%

Key: n = number of subjects.

**Table 2 jpm-14-00372-t002:** IT-MAIS questionnaire assessment from control group and intervention group for pre- and post-intervention period in children aged between 6 and 8 years, 2023.

IT-MAIS Questionnaire	Control Group				Intervention Group			
	Mean	95% CI		*p* *	Mean	95% CI		*p* *
**Pre-intervention**	17.3	9.6	25.0	**0.01**	27.4	21.0	33.0	**0.16**
**Post-intervention**	26.8	21.0	32.0		29.6	24.0	34.0	

* *p*-value obtained via Student’s *t*-test, for paired measures, significant if <0.05.

**Table 3 jpm-14-00372-t003:** TFDF protocol assessment from control group and intervention group for the pre- and post-intervention period in children aged between 6 and 8 years, 2023.

TFDF	Control Group				Intervention Group			
	Mean	95% CI		* p * *	Mean	95% CI		* p * *
Pre-intervention	24	16	32	0.07	23	18	27	0.01
Post-intervention	27	19	36		28	21	34	

* *p*-value obtained via Student’s *t*-test, for paired measures, significant if <0.05.

## Data Availability

The data is available in this article.

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
