# Peer review of "Hearing and Language Skills in Children Using Hearing Aids: Experimental Intervention Study"

_jpm, 2024, doi:10.3390/jpm14040372_

Round 1

Reviewer 1 Report

Comments and Suggestions for Authors

Dear Author,

Good attempt to build evidence based practice information for importance of auditory training. I think in your introduction, and justification for the study can be more emphasised on this point. Following information is good to bring to the notice of the readers.

1. Hearing loss age of the children with hearing impairment

2. What type of personal amplification devices were used. At lest weather they are analog or digital/ body level vs ear level. information. Duration of usage (at least as per the report from parents)

3. Aided thresholds. Functional gain or insertion gain information will add value. This proves, that all participants had adequate aided thresholds to perceive speech and language stimulation. 

4. If you have reliable data on the above, a comparison and or qualitative statement on the relation between these and improvement can be good. 

Author Response

The requests were met and are described in blue in the manuscript text

Reviewer 2 Report

Comments and Suggestions for Authors

1.     Figure 1-3 are not clear, please revise.

2.     The layouts of table 1-3 are inappropriate. Please revise.

3.     Please revise the format and citation of references to those of MDPI.

4.     Conclusion section (Line 336-349) should be more concise. 

Comments on the Quality of English Language

Moderate editing of English language required.

Author Response

1.     Figure 1-3 are not clear, please revise.

adjusted

2.     The layouts of table 1-3 are inappropriate. Please revise

adjusted

3.     Please revise the format and citation of references to those of MDPI.

adjusted

4.     Conclusion section (Line 336-349) should be more concise.

adjusted

Reviewer 3 Report

Comments and Suggestions for Authors

Dear Ladies and Gentlemen, Dear Journal-Team,

the interesting manuscript 'Hearing and language skills in children using hearing aids: experimantal interevention study' underlines the importance of hearing skills improvement in children and gives hints for the difficult interventional decision-making in children. Please mention the hearing characteristics of the control group and the intervention group in more detail, and compare when necessary in a table, to facilitate decision-making for the type of intervention. The manuscript is well written.

1. Please check the birth period in the Results section: 2015-2016?

2. Are in Figure 1 five or six sessions explained? Please change to English in Figure 2 and Figure 3. Table 1 is a kind of big formatted.

3. Please check the references for accuracy according to the Journal Style Guidelines. Use English for reference accessability in reference 1, 11 and 12 (American Speech and Hearing Association, The Joint Committee on Infant Hearing, Yoshida et al.). Give an English translation for references 3, 5, 6, 10, 12 (Beier et al., Cruzatti et al., Oliveira et al., Souza, Yoshida) and mention that the reference is in Portugese at the end of the reference. Check for uniform number in mentioning the autor names (reference 5 by Cruzatti et al., reference 13 by Yoshinga-Itano et al.). Check for uniform writing style of the Journal title. For Yoshinga-Itano et al., the publisher location is mentioned in contrast to the other journal references.

Sincerely, 

Author Response

Commentary revisor 3

Modification

1. Please check the birth period in the Results section: 2015-2016?

adjusted

2. Are in Figure 1 five or six sessions explained? Please change to English in Figure 2 and Figure 3. Table 1 is a kind of big formatted.

adjusted

3. Please check the references for accuracy according to the Journal Style Guidelines. Use English for reference accessability in reference 1, 11 and 12 (American Speech and Hearing Association, The Joint Committee on Infant Hearing, Yoshida et al.). Give an English translation for references 3, 5, 6, 10, 12 (Beier et al., Cruzatti et al., Oliveira et al., Souza, Yoshida) and mention that the reference is in Portugese at the end of the reference. Check for uniform number in mentioning the autor names (reference 5 by Cruzatti et al., reference 13 by Yoshinga-Itano et al.). Check for uniform writing style of the Journal title. For Yoshinga-Itano et al., the publisher location is mentioned in contrast to the other journal references.

adjusted

Round 2

Reviewer 2 Report

Comments and Suggestions for Authors

The authors address my concerns. 

Thanks

Comments on the Quality of English Language

Minor editing of English language required.